# LLMs are Single-threaded Reasoners: Demystifying the Working Mechanism of Soft Thinking

**Junhong Wu,*Jinliang Lu, Zixuan Ren, Gangqiang Hu, Zhi Wu, Dai Dai,† Hua Wu**

Baidu Inc., Beijing, China

## Abstract

Human cognition naturally engages with abstract and fluid concepts, whereas existing reasoning models often rely on generating discrete tokens, potentially constraining their expressive capabilities. Recent advancements aim to address this limitation by enabling large language models (LLMs) to generate soft, abstract tokens, thus facilitating reasoning within a continuous concept space. In this paper, we investigate the *Soft Thinking* capabilities of various LLMs through a systematic analysis of their internal behavior using a suite of probing techniques. Contrary to the prevailing belief that Soft Thinking supports parallel exploration of diverse reasoning paths, our findings reveal that **LLMs behave as single-threaded reasoners**—they predominantly rely on the token with the highest probability in the soft input to predict the next step. This behavior induces a greedy feedback loop that suppresses alternative reasoning paths and undermines the benefits of transmitting richer information via Soft Tokens. To address this *Greedy Pitfall*, we propose **Stochastic Soft Thinking**, which introduces stochasticity to break free from this Greedy Pitfall. Our experiments demonstrate that incorporating *randomness*—particularly with the **Gumbel-Softmax trick**—can alleviate the limitations of vanilla approaches and unleash the potential of Soft Thinking, resulting in superior performance across eight reasoning benchmarks. We further demonstrate that *Stochastic Soft Thinking* exhibits stronger exploration potential compared to conventional COT. Our findings deepen the understanding of continuous reasoning and establish the foundation for future work on improving Soft Thinking with Reinforcement Learning.

## 1 Introduction

Large Language Models (LLMs) have achieved remarkable progress across a wide range of tasks (Jaech et al., 2024; DeepSeek-AI et al., 2025). A key driver of this success is *Chain-of-Thought* (CoT) prompting, which enables models to solve complex problems by generating intermediate reasoning steps in natural language (Kojima et al., 2022). However, CoT inherently constrains the reasoning process to sequences of discrete tokens, which may limit the model's ability to explore alternative solutions and reason beyond the bounds of natural language (Yao et al., 2023).

In contrast, neuroscientific studies suggest that many aspects of human reasoning operate independently of language, engaging distinct brain regions (Fedorenko & Varley, 2016; Benn et al., 2023; Fedorenko et al., 2024). Inspired by this insight, recent work has proposed replaceing discrete tokens with continuous representations such as hidden states or output distributions for decoding (Hao et al., 2024; Shen et al., 2025; Zhang et al., 2025). These approaches aim to enable LLMs to reason with abstract, high-bandwidth signals and explore multiple reasoning paths in parallel (Hao et al., 2024; Zhang et al., 2025; Gozeten et al., 2025; Zhu et al., 2025).

Despite promising results, the underlying mechanisms of *Soft Thinking* remain poorly understood. In this paper, we conduct a systematic investigation into the behavior of Soft Thinking in modern

---

*Contact: 1600013522@pku.edu.cn, daidai@baidu.com
†Corresponding author

LLMs with strong reasoning capabilities (DeepSeek-AI et al., 2025; Team, 2025; He et al., 2025). Surprisingly, our preliminary experiments reveal that the vanilla Soft Thinking approach performs significantly worse than conventional token sampling-based decoding methods. To understand this discrepancy, we analyze the model's internal dynamics using a suite of probing techniques. Our findings challenge the prevailing assumption that Soft Thinking facilitates parallel exploration of multiple reasoning paths. As illustrated in Fig 1, LLMs tend to rely predominantly on the single token with the highest probability in the soft input to predict the next step. This behavior induces a feedback loop that reinforces the most confident reasoning trajectory, thereby suppressing the exploration of alternative paths. While Soft Tokens do transmit richer information and can potentially unlock novel reasoning trajectories, this *Greedy Pitfall* prevents the LLM from fully capitalizing on this advantage.

To address this issue, we introduce *Stochastic Soft Thinking*, a framework that restores controlled randomness into the reasoning process using techniques such as Dirichlet sampling and the Gumbel-Softmax trick (Gumbel, 1954; Maddison et al., 2014; Kool et al., 2019). Our empirical evaluations on challenging reasoning benchmarks across three mainstream LLMs show that these methods mitigate greedy behavior and unlock the potential of Soft Thinking. In particular, Gumbel-Softmax offers a flexible trade-off between randomness and smoothness, leading to consistent performance improvements over both vanilla Soft Thinking and traditional CoT. With controllable stochasticity, we enable Test-time Scaling for Soft Thinking and observed stronger exploration potential compared to standard Token CoT. Our findings deepen the understanding of continuous reasoning and establish the foundation for future work on improving Soft Thinking with Reinforcement Learning.

**Our contributions are threefold:**

- We present the first in-depth analysis of Soft Thinking in LLMs, revealing that it does not inherently support parallel reasoning and is dominated by top-1 token signals.

- We propose *Stochastic Soft Thinking*, demonstrating that controlled randomness—especially via Gumbel-Softmax—significantly improves reasoning performance.

- We demonstrate that *Stochastic Soft Thinking* exhibits stronger exploration potential compared to conventional COT, establishing the foundation for future work on improving Soft Thinking with Reinforcement Learning.

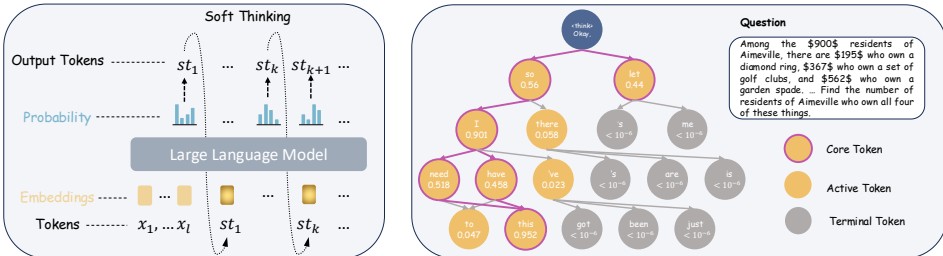

Figure 1: **Left:** Soft Thinking replaces the discrete token $t$ with the Soft Token $st$ (defined as the probability distribution over vocabulary). **Right:** Soft Thinking predominantly explores branches associated with the top-1 token. In contrast, paths stemming from non-top-1 tokens are typically terminated in the next step.

## 2 BACKGROUNDS: DISCRETE THINKING AND SOFT THINKING

### 2.1 DISCRETE THINKING

We begin by outlining the discrete token generation process in LLMs. Given a query $\mathbf{x}$, the model generates a sequence $\mathbf{y}$, which includes both the intermediate reasoning path $\mathbf{t}$ and the final answer $\mathbf{a}$. Initially, the model generates the intermediate reasoning path $\mathbf{t}$. Upon decoding a special end-of-thinking token, $<$EOT$>$, the model transitions to generating the final answer $\mathbf{a}$. During each decoding step, the LLM uses the input $\mathbf{x}$ and all previously generated tokens $\mathbf{y}_{<t}$ as context to

predict the next token $\mathbf{y}_t$:

$$\mathbf{y}_t \sim p = \text{LLM}(\mathbf{x}, \mathbf{y}_{<t}) \in \Delta^{|V|-1} \tag{1}$$

Here, $V$ represents the vocabulary of the LLM, and $\Delta^{|V|-1}$ is the probability simplex over the vocabulary, representing all possible probability distributions.

## 2.2 SOFT THINKING

In the discrete thinking process, the selection of only one token inherently restricts the expressive capacity of LLMs to natural language and may result in potential information loss. To address this limitation, a more intuitive approach is to bypass the final token selection process entirely, instead feeding the entire probability distribution into the next step of the model (Zhang et al., 2025). To formally define this vanilla Soft Thinking process, we introduce the following terms.

**Definition 1** (*Soft Token*). The soft token is defined as the generated probability distribution over the LLM's vocabulary:

$$\mathbf{st} := p = \text{LLM}(\mathbf{x}, \mathbf{y}_{<t}) \in \Delta^{|V|-1} \tag{2}$$

**Definition 2** (*Soft Input*). To feed a *Soft Token* into the model, we employ the embedding matrix of the model $\mathbf{E} \in R^{|V| \times d}$. Here, $d$ represents the embedding size of the model. Let $\mathbf{e}_i$ denote the embedding vector of the token $\mathbf{t}_i$. The embedding of a Soft Token, denoted as $\mathbf{E}(st)$, is calculated as the weighted sum of the individual token embeddings:

$$\mathbf{E}(st) := \sum_{k=1}^{|V|} p_i \cdot \mathbf{e}_i \tag{3}$$

This construction ensures $\mathbf{E}(st)$ is a convex combination of all embedding vectors. As a result, it remains within the manifold of the original LLM input space, which helps alleviate the issue of out-of-distribution (OOD) inputs (Yuan et al., 2023). Preserving all tokens in the vocabulary can introduce noise and significantly increase computational overhead. To address this, we employ top-$k$ or top-$p$ truncation followed by renormalization, effectively truncating the unreliable tail of the probability distribution (Holtzman et al., 2020).

**Soft Thinking Process**. In practice, Soft Thinking is applied only during the intermediate reasoning process, replacing the discrete token CoT. At each step, the model outputs a *Soft Token* and constructs its corresponding *Soft Input* for the next generation step. When the top-1 token of the generated *Soft Token* is , the reasoning process is terminated, and the model switches to discrete token decoding to generate a readable final answer.

## 2.3 POSSIBLE WORKING MECHANISM OF SOFT THINKING

One of the most intriguing properties of Soft Thinking is its potential to inherently preserve multiple reasoning paths, effectively forming a latent search tree. COCONUT suggested that models utilizing Soft Thinking can maintain a diverse set of possibilities within the continuous reasoning process (Hao et al., 2024). Many works (Zhang et al., 2025; Zhu et al., 2025; Gozeten et al., 2025) propose theoretical frameworks demonstrating how Soft Thinking can perform implicit parallel search. However, the empirical evidence supporting these hypotheses is still absent.

## 3 PRELIMINARY EXPERIMENTS: IS SOFT THINKING EFFECTIVE?

In this section, we present our initial experiments with a vanilla implementation of the Soft Thinking approach. We begin by detailing the tasks and LLMs employed in our experiments. Subsequently, we demonstrate that vanilla Soft Thinking generally underperforms compared to token CoT.

### 3.1 EXPERIMENT SETTING

**Models.** We utilize a range of mainstream LLMs known for their reasoning capabilities. These include Deepseek-R1-Distill-Qwen-32B (DeepSeek-AI et al., 2025), QwQ-32B (Team, 2025), and Skywork-OR1-32B (He et al., 2025).

**Benchmarks.** Our evaluation is conducted on eight diverse benchmarks. These include AIME'24/'25 (Mathematical Association of America, 2024), MATH-500 (Cobbe et al., 2021), and AMC'23 mathematics competitions. (2023) in the mathematical domain, GPQA-Diamond (Rein et al., 2023) for knowledge-based question-and-answer, and HumanEval (Chen et al., 2021), MBPP (Austin et al., 2021), and LiveCodeBench (Jain et al., 2025) for code generation. This comprehensive suite allows us to thoroughly assess the performance across varied domains and tasks. Please refer to Appendix A for details.

**Implementation Details.** We utilize the SGLang (Zheng et al., 2024) Soft Thinking implementation provided in Zhang et al. (2025). For both vanilla Soft Thinking and discrete Token Thinking, we set the temperature to 0.6, top-$p$ to 0.95, and top-$k$ to 30. Across all experiments, the maximum generation length is capped at 32,768 tokens. For the AIME'24/'25 and AMC'23 benchmarks, we report Average@32 to mitigate the effects of randomness in smaller test sets. For the remaining benchmarks, we report Pass@1.

Table 1: Detailed results on math, knowledge Q&A, and code benchmarks. Results highlighted in green indicate an improvement or performance comparable to Token CoT. Results highlighted in red signal a decline in performance relative to Token CoT.

| Thinking Mode | AIME24 | AIME25 | MATH500 | AMC23 | GPQA-Diamond | HumanEval | MBPP | LiveCodeBench | Avg |
|---|---|---|---|---|---|---|---|---|---|
| Deepseek-R1-Distill-Qwen-32B | | | | | | | | | |
| Token (Greedy) | 66.66 | 50.00 | 92.20 | 85.00 | 60.10 | 87.20 | 88.71 | 42.65 | 71.57 |
| Token (Sampling) | 72.08 | 55.63 | 94.50 | 95.46 | 60.60 | 97.25 | 95.13 | 57.35 | 78.50 |
| Soft (Vanilla) | 62.00 | 49.17 | 91.60 | 90.00 | 60.10 | 86.41 | 87.93 | 44.80 | 71.50 |
| QwQ-32B | | | | | | | | | |
| Token (Greedy) | 80.00 | 70.00 | 97.00 | 100.00 | 64.14 | 95.12 | 96.10 | 58.78 | 82.64 |
| Token (Sampling) | 77.92 | 67.50 | 96.20 | 97.50 | 62.63 | 98.17 | 96.89 | 62.00 | 82.35 |
| Soft (Vanilla) | 76.67 | 62.29 | 96.20 | 98.75 | 59.60 | 93.90 | 95.33 | 57.71 | 80.06 |
| Skywork-OR1-32B | | | | | | | | | |
| Token (Greedy) | 76.67 | 73.33 | 95.80 | 90.00 | 56.06 | 81.71 | 86.38 | 54.84 | 76.85 |
| Token (Sampling) | 78.75 | 71.25 | 96.40 | 98.28 | 62.62 | 96.95 | 97.28 | 62.37 | 82.99 |
| Soft (Vanilla) | 79.16 | 69.38 | 96.00 | 97.97 | 59.60 | 85.37 | 90.66 | 55.56 | 79.21 |

## 3.2 EXPERIMENTAL RESULTS

As shown in Table 1, across all tested large language models (LLMs), vanilla Soft Thinking generally underperforms compared to discrete Token Thinking. Overall, we find that vanilla Soft Thinking does not lead to improvements compared to discrete Token thinking in accuracy. Instead, it appears to achieve similar performance with greedy decoding.

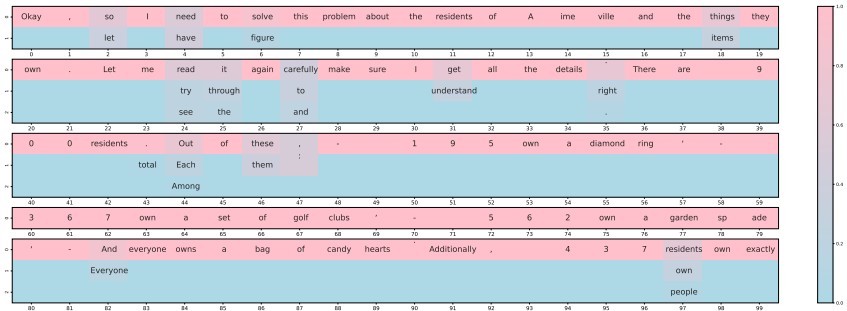

Figure 2: An example illustrating the probability distribution of the vanilla Soft Thinking method.

## 4 ANALYSIS OF SOFT THINKING BEHAVIOR

Our preliminary experiments revealed that its performance did not surpass that of discrete token decoding methods. This unexpected result prompted us to question the actual behavior of Soft Thinking in practice: *Why does incorporating vanilla Soft Thinking not lead to the anticipated performance improvements?*

## 4.1 CASE STUDY

Figure 2 presents a case study of the vanilla Soft Thinking process applied to a simple question. Through this example, we observe that, in consecutive decoding steps, the components of the subsequent token consistently and exclusively exhibit semantic coherence with the dominant token from the preceding step. For instance, in steps 2 and 3, the word 'I' is not a semantically appropriate successor to 'let'. Based on these observations, we propose the following hypothesis:

> **Hypothesis: LLMs are Signle-Threaded Reasoners**
>
> LLMs lack the ability to process multiple different semantic trajectories in parallel. When a Soft Token is fed into an LLM, the generation process is typically dominated by the majority component of the Soft Token.

To verify this hypothesis, we employ QwQ-32B to generate responses for AIME'24 and AIME'25. We collect prediction distributions for nearly $10^6$ steps and only preserve the intermediate reasoning steps. Subsequently, we analyze the decoding behavior of these generated responses.

## 4.2 OUTPUT PROBABILITY OF SOFT THINKING

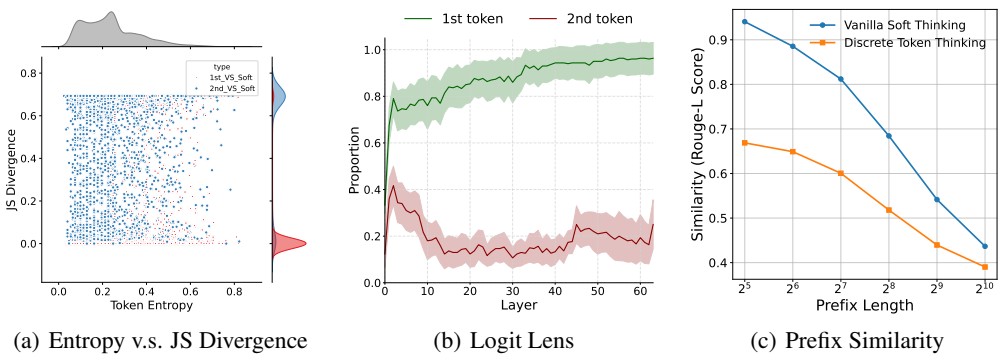

| (a) Entropy v.s. JS Divergence | (b) Logit Lens | (c) Prefix Similarity |

Figure 3: Output entropy/token probability vs. JS-divergence between next token prediction probabilities yield from different inputs. The prediction of soft input is nearly identical to the prediction of the 1st token input, but completely different from the prediction of the 2nd token input.

We begin by examining the model's output probabilities during identified soft steps by conducting three separate forward passes and observing their outcomes: one pass uses the entire Soft Token, denoted as $P_{\mathbf{st}}$, while the other two use the highest and second-highest probability tokens within the Soft Token, denoted as $P_1$ and $P_2$, respectively. We employ Jensen-Shannon (JS) Divergence to measure the variation in prediction behavior resulting from these distinct inputs.

As shown in Figure 3(a), the JS divergence between the Soft Thinking result, $P_{\mathbf{st}}$, and the highest probability token thinking result, $P_1$, shows a highly concentrated distribution around 0. This suggests that the model's prediction during a Soft Thinking step closely matches the outcome with just the specific token, implicating a significant influence of the top-1 token on the model's reasoning process. In contrast, the JS divergence between the Soft Thinking result, $P_{\mathbf{st}}$, and the second-highest probability token thinking result, $P_2$, frequently approaches the maximum value of JS divergence, indicating that the second-highest probability token has limited influence on the thinking process.

## 4.3 DECODING HIDDEN STATES BY LOGIT LENS

To deepen our understanding of the decoding process, we apply the Logit Lens technique (nostalgebraist, 2020) to track the reasoning paths of various components of the Soft Token. This method involves normalizing the output of intermediate hidden states and projecting them through the LM head. Previous studies have shown that these projections often yield interpretable, top-ranked tokens that align closely with the model's intermediate representations (Dar et al., 2023; Geva et al., 2023).

We start by pinpointing branching points where the model generates a Soft Token consisting of at least two semantically diverse tokens[1]. To construct a balanced Soft Token, we manually assign probabilities of 0.6 to the first token and 0.4 to the second. Following the previous section, we perform three separate forward passes and apply the Logit Lens. For the Soft Token forward pass, we extract the top 10 tokens, while for the two single-token forward passes, we extract the top 5 tokens each. We then plot the size of the intersection between the Logit Lens results from different forward passes to illustrate the extent of each token's reasoning path within the Soft Token process.

As depicted in Figure 3(b), the representation of both single tokens' reasoning paths gradually rises within the first 2-3 layers. This suggests that the model initially considers both reasoning paths in parallel. However, as processing continues through additional layers, the prominence of the first token's path steadily increases to 1.0, while the second token's path decreases[2]. This pattern indicates that the forward process inherently acts as a pruner, progressively favoring the reasoning path of the first token while diminishing that of the second.

### 4.4 GREEDY PITFALL

As demonstrated in previous sections, LLMs predominantly rely on the top-1 token for forward computation. This tendency creates a positive feedback loop where the model becomes entrenched in its most self-assured reasoning path—a phenomenon we term the "Greedy Pitfall." To further substantiate this, we evaluate the sequence similarity between vanilla Soft Thinking, discrete Token Thinking, and Greedy Token Thinking. Specifically, we concatenate the top-1 token from each step in vanilla Soft Thinking to form a coherent reasoning trace and compute the ROUGE-L metric using the Greedy Token Thinking trace as the reference. As shown in Figure 3(c), the ROUGE-L score of vanilla Soft Thinking is significantly higher than that of discrete Token Thinking, indicating that vanilla Soft Thinking inherently exhibits greedy behavior. This explains why vanilla Soft Thinking underperforms, as the reasoning path that maximizes likelihood is proven to be generic, repetitive, and awkward (Holtzman et al., 2020).

## 5 UNLEASHING THE POTENTIAL OF SOFT THINKING WITH RANDOMNESS

Our experiments and analysis reveal that vanilla Soft Thinking tends to be inherently greedy and does not necessarily enhance performance. However, this does not entirely negate the potential of Soft Thinking. Specifically, as illustrated in Figure 3(a), there are instances where the richer information in Soft Token leads to reasoning branches that diverge from the 1st token's decoding path. As the generation process continues, these branches cause the model to gradually deviate from the greedy trace, as evidenced by the decreasing ROUGE-L scores shown in Figure 3(c). Despite this potential for stronger expressive capabilities, the overall tendency towards greediness obscures the benefits of the soft input, leading to suboptimal performance. To uncover the true potential of Soft Thinking, we are exploring approaches to break free from the "Greedy Pitfall."

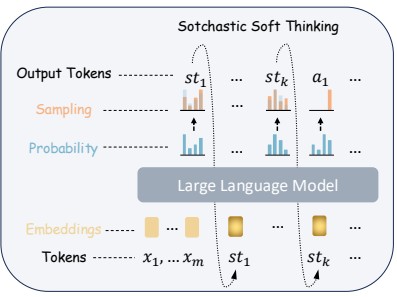

Figure 4: An illustration for Stochastic Soft Thinking, which incorporates random sampling techniques to construct a Stochastic Soft Token.

### 5.1 STOCHASTIC SOFT THINKING

Previous works have established that the top-quality generations obtained from the model rely on *randomness* in the decoding method (Holtzman et al., 2020). We believe that such randomness is similarly crucial for effective Soft Thinking and further RL training (Appendix **??**). Therefore, we explore strategies to introduce randomness into the Soft Thinking process. Intuitively, our focus

---

[1]We select semantically diverse Soft Thinking steps according to the JS divergence between the first and the second token's forward results.

[2]The prominence of the second token does not converge to zero, because it shares some tokens with reasoning paths of the first token.

is on generating Stochastic Soft Tokens with controllable randomness, $st'$, as shown in Fig 4. We begin by outlining several key properties that a Stochastic Soft Token $st'$ should possess:

- **Validness**: It should still be a *valid probability distribution* over $V$.

- **Randomness**: It should be *unbiased* and reflect *the original predictive information* in $st$.

- **Softness**: It should be *soft* and do not collapse into a one-hot vector.

Guided by these principles, we explore two different approaches, namely, Dirichlet Sampling and the Gumbel-Softmax trick.

### 5.1.1 DIRICHLET SAMPLING

Perhaps the most common distribution over the probability simplex $\Delta^{n-1}$ is the Dirichlet distribution. Intuitively, we could set the model's output distribution $p$ as the concentration parameters and construct a corresponding Dirichlet distribution $\mathrm{Dir}(p)$. However, directly using $p$ as concentration parameters will cause the probability mass to concentrate near the simplex corners, resulting in nearly one-hot vectors. To this end, we introduce a scaling parameter $\gamma$ and sample from $\mathrm{Dir}(\gamma p)$.

### 5.1.2 GUMBEL-SOFTMAX TRICK

The Gumbel-Max trick is an algorithm for sampling from a categorical distribution over classes $i \in 1, ..., n$ with probability $\pi$. The algorithm first samples independent noises $g_i$ from the Gumbel distribution $G(0, 1)$. Subsequently, the sampled noise is applied to the logits $\log(\pi_i)$ of probability $\pi$, followed by an $\mathrm{argmax}$ operation. The Gumbel-Softmax distribution utilizes the softmax function with temperature $\tau$ as a continuous, differentiable approximation to $\mathrm{argmax}$, deriving $n$-dimensional sample vectors $y \in \Delta^{n-1}$, where:

$$y_i = \frac{\exp\left((g_i + \log(\pi_i))/\tau\right)}{\sum_{k=1}^{n} \exp\left((g_k + \log(\pi_k))/\tau\right)} \tag{4}$$

Empirically, the Gumbel-Softmax distribution interpolates between discrete one-hot-encoded categorical distributions and continuous categorical densities (Maddison et al., 2014; Kool et al., 2019). Therefore, applying this trick to the original Soft Token yields a valid Stochastic Soft Token.

## 5.2 EXPERIMENTAL RESULTS

Table 2: Detailed results on math, knowledge Q&A, and code benchmarks. Results highlighted in green indicate an improvement or performance comparable to Token CoT. Results highlighted in red signal a decline in performance relative to Token CoT

|  | AIME24 | AIME25 | MATH500 | AMC23 | GPQA-Diamond | HumanEval | MBPP | LiveCodeBench | Avg |
|---|---|---|---|---|---|---|---|---|---|
| | | | | | Deepseek-R1-Distill-Qwen-32B | | | | |
| Token (Sampling) | 72.08 | 55.63 | 94.50 | 95.46 | 60.60 | 97.25 | 95.13 | 57.35 | 78.50 |
| Soft (Vanilla) | 62.00 | 49.17 | 91.60 | 90.00 | 60.10 | 86.41 | 87.93 | 44.80 | 71.50 |
| Soft (Dirichlet) | 69.79 | 54.58 | 94.60 | 94.53 | 62.12 | 98.17 | 95.72 | 57.35 | 78.36 |
| Soft (Gumbel) | 72.92 | 55.42 | 96.00 | 95.62 | 63.13 | 98.17 | 95.64 | 59.50 | 79.55 |
| | | | | | QwQ-32B | | | | |
| Token (Sampling) | 77.92 | 67.5 | 96.20 | 97.5 | 62.63 | 98.17 | 96.89 | 62.00 | 82.35 |
| Soft (Vanilla) | 76.67 | 62.29 | 96.20 | 98.75 | 59.60 | 93.90 | 95.33 | 57.71 | 80.06 |
| Soft (Dirichlet) | 76.67 | 68.13 | 96.60 | 96.56 | 61.62 | 96.34 | 95.72 | 59.50 | 81.39 |
| Soft (Gumbel) | 78.96 | 68.95 | 97.20 | 98.28 | 67.67 | 97.56 | 97.66 | 62.72 | 83.63 |
| | | | | | Skywork-OR1-32B | | | | |
| Token (Sampling) | 78.75 | 71.25 | 96.40 | 98.28 | 62.62 | 96.95 | 97.28 | 62.37 | 82.99 |
| Soft (Vanilla) | 79.16 | 69.38 | 96.00 | 97.97 | 59.60 | 85.37 | 90.66 | 55.56 | 79.21 |
| Soft (Dirichlet) | 78.96 | 71.25 | 96.20 | 97.50 | 66.16 | 96.34 | 97.28 | 61.29 | 83.12 |
| Soft (Gumbel) | 79.79 | 73.75 | 97.40 | 98.59 | 67.67 | 97.56 | 98.05 | 64.16 | 84.62 |

We implemented two randomization approaches within the Soft Thinking framework introduced in Section 2 and conducted experiments adhering to the protocol outlined in Section 3.1. For Dirichlet Sampling, the scaling parameter $\alpha$ was tested in the range of $[1.0, 10.0]$ with increments of $1.0$. For the Gumbel-Softmax trick, the temperature hyperparameter $\tau$ was tested in the range of $[0.3, 0.9]$ with increments of $0.1$. We found that setting $\alpha = 4.0$ and $\tau = 0.5$ yielded good performance, and these values were used as the default hyperparameters. The results across eight benchmarks are presented in Table 2.

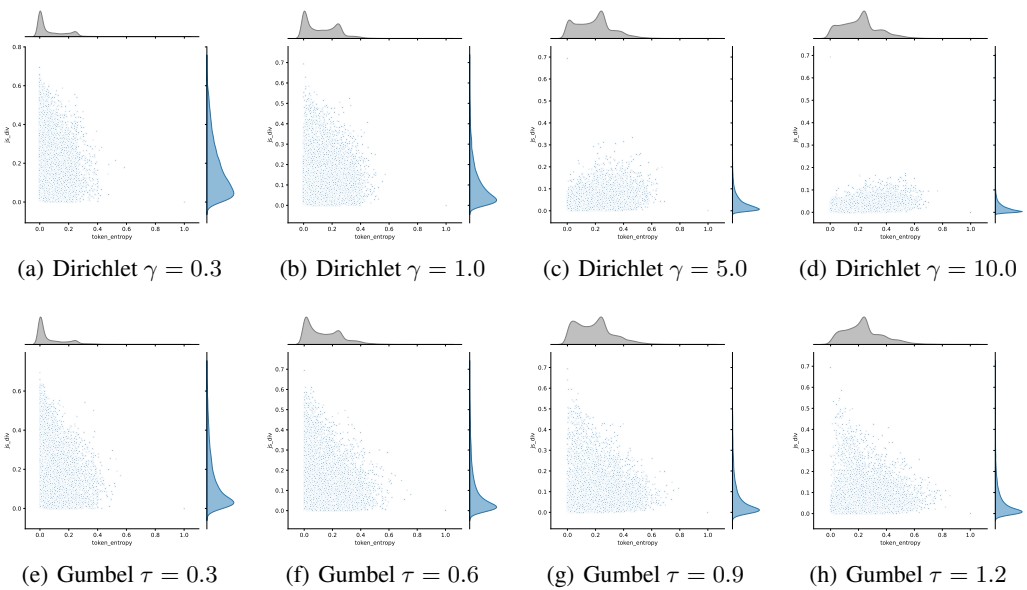

(a) Dirichlet $\gamma = 0.3$  (b) Dirichlet $\gamma = 1.0$  (c) Dirichlet $\gamma = 5.0$  (d) Dirichlet $\gamma = 10.0$

(e) Gumbel $\tau = 0.3$  (f) Gumbel $\tau = 0.6$  (g) Gumbel $\tau = 0.9$  (h) Gumbel $\tau = 1.2$

Figure 5: Softness vs. randomness for Stochastic Soft Tokens.

Both Stochastic Soft Thinking approaches demonstrated improved performance compared to vanilla Soft Thinking. This suggests that incorporating randomness effectively mitigates the "Greedy Pitfall." Notably, only the approach utilizing the Gumbel-Softmax trick achieved performance gains beyond those of discrete Token Thinking. We conducted further experiments to understand this difference.

## 5.3 BALANCING RANDOMNESS AND SOFTNESS

To understand the performance differences between various Stochastic Soft Thinking approaches, we quantified the randomness and softness of each method. Specifically, we used the normalized entropy of generated Soft Tokens as a proxy for softness. For randomness, we calculated the JS divergence between the original LLM output probability $\pi$ and the Stochastic Soft Token $st'$.

As illustrated in Figure 5, the Gumbel-Softmax trick allows for easy adjustment of the softness by varying the temperature hyperparameter $\tau$, while consistently maintaining adequate randomness, indicated by high JS divergence. In contrast, Dirichlet sampling struggles to balance randomness and softness. When the scaling parameter $\gamma \to 1$, the Stochastic Soft Token exhibits sufficient randomness but collapses into a near one-hot vector, as indicated by the low entropy. Conversely, when the scaling parameter $\gamma$ increases, the Stochastic Soft Token gradually converges to the original model's output probability distribution, displaying higher softness but limited randomness. This difference highlights that the Gumbel-Softmax trick can effectively leverage both the advantages of randomization and Soft Tokens, whereas the Dirichlet approach cannot.

## 5.4 THEORETICAL JUSTIFICATION: GUMBEL-SOFTMAX TRICK AND LUCE'S CHOICE AXIOM

The Gumbel-Softmax trick is not merely an experimentally convenient method for sampling from categorical distributions; it is theoretically well-motivated for constructing Stochastic Soft Tokens because it uniquely satisfies Luce's choice axiom (Luce et al., 1959). Luce's choice axiom asserts that the probability of choosing an item from a set depends solely on its utility relative to the sum of all utilities in the set, independent of other alternatives. This property is crucial for ensuring that selection probabilities reflect the true relative preferences among items. The Gumbel-Softmax distribution naturally satisfies Luce's axiom (Maddison et al., 2014; Kool et al., 2019). By adding Gumbel noise to the logarithm of the utilities (or probabilities), the Gumbel-Max trick ensures that

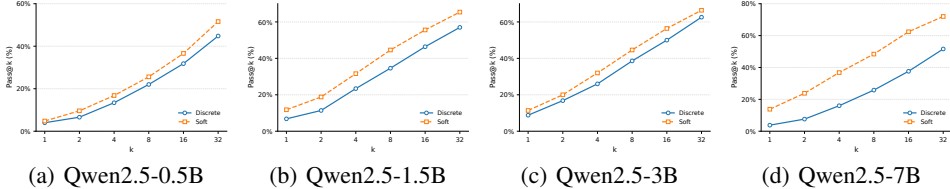

(a) Qwen2.5-0.5B     (b) Qwen2.5-1.5B     (c) Qwen2.5-3B     (d) Qwen2.5-7B

Figure 6: Pass@k Comparison.

the selection probabilities are proportional to the original utilities:

$$\underset{i}{\operatorname{argmax}}[g_i + \log(\pi_i)] \sim \frac{\exp(\log(\pi_i))}{\sum_i \exp(\log(\pi_i))} = \pi_i \tag{5}$$

We can generalize the $\operatorname{argmax}$ operation to a $\operatorname{argtopk}$ operation, which returns the indices of the k largest values, in order of decreasing value. Kool et al. (2019) proved the following theorem:

**Theorem 1.** For $k \leq n$, let $I^1, ..., I^k$ be $\operatorname{argtopk}_i [g_i + \log(\pi_i)]$. Then $I^1, ..., I^k$ is an (ordered) sample without replacement from the Categorical Distribution $\pi$ on a pool $N$. This means, for a realization $i^1, ..., i^k$, the following holds:

$$P(I^1 = i^1, ..., I^k = i^k) = \prod_{j=1}^{k} \frac{\pi_{i^j}}{\sum_{N_j} \pi_{i^j}} \tag{6}$$

where $N_j = N \setminus \{i^1, ..., i^{j-1}\}$ is the domain (without replacement) for the $j$-th sampled element.

This theorem shows that the Gumbel-Max trick naturally extends to sampling multiple items without replacement while preserving proportional probabilities. Such a mechanism closely mirrors the ideal process of constructing Stochastic Soft Tokens from LLM output distributions, where tokens are sequentially selected according to their relative utilities among the remaining options. Replacing the $\operatorname{argtopk}$ with $\operatorname{Softmax}$ and top-$k$ renormalization provides a relaxation: the discrete ranking is converted into a continuous distribution, whose probabilities preserve the ranking information and can be directly used to weight token embeddings when constructing the input for the next generation step.

### 5.5 STRONGER EXPLORATION POTENTIAL

To further demonstrate the potential of Stochastic Soft Thinking in reinforcement learning (RL), we evaluate the Pass@k metric as introduced by Brown et al. (2024) across various base models. Pass@k measures the proportion of problems a model can potentially solve within k attempts, and is commonly used to assess the capability boundaries of large language models (LLMs) (Yue et al., 2025; Chen et al., 2025). A higher Pass@k score indicates stronger exploration ability, which is critical for effective RL training.

Figure 6 presents the Pass@k performance ($k = 1, 2, 4, 8, 16, 32$) on the MATH500 benchmark for the Qwen2.5 base model, ranging from 0.5B to 7B parameters. As shown, Stochastic Soft Thinking consistently outperforms Discrete Token Thinking across all settings, suggesting that incorporating Stochastic Soft Thinking into rollouts and RL training may be a highly promising direction. Due to time and resource constraints, we leave a full RL integration for future work.

## 6 LIMITATION AND FUTURE WORKS

While our research primarily focuses on a training-free Soft Thinking approach, our findings also shed light on fine-tuning LLMs to support this method of reasoning. Notably, we discovered that LLMs typically function as single-threaded reasoners, lacking the ability for parallel reasoning. This limitation appears to stem from an inductive bias inherent in both the Transformer architecture and the next-token prediction objective used in pretraining and fine-tuning. Due to this strong inductive

bias, enabling LLMs to engage in parallel reasoning through Soft Thinking presents significant challenges.

Supervised fine-tuning can lead to a substantial distributional shift, resulting in catastrophic forgetting of the knowledge acquired during pre-training. This issue may explain why approaches like COCONUT (Hao et al., 2024) and its variants struggle to match the performance of token-based CoT, as they require fine-tuning the model to compress multiple tokens into one embedding. Although recent studies show that reinforcement learning (RL) can enhance reasoning abilities without forgetting (Lai et al., 2025), it is also constrained by the capabilities of the base model (Yue et al., 2025).

As we have shown in the paper, existing LLMs are unable to generate Soft Thinking traces with parallel reasoning characteristics. Therefore, relying solely on RL to facilitate parallel reasoning may not be a feasible approach. Valuable directions include exploring multi-token prediction pre-training and novel architectures to build the ability of parallel thinking. Nonetheless, our experiments indicate that Soft Thinking demonstrates certain advantages beyond parallel thinking, attributed to the enriched information conveyed through Soft Tokens. Future research could concentrate on harnessing this potential to improve the reasoning capabilities of LLMs.

## 7 RELATED WORK

Chain-of-thought (CoT) prompting enhances language model performance by facilitating step-by-step reasoning through natural language (Kojima et al., 2022). However, this approach can be inefficient due to its reliance on discrete text tokens. To address this, various previous studies have investigated the potential of reasoning in a continuous space. Yang et al. (2024) and Shalev et al. (2024) investigate the implicit reasoning capabilities of transformers with multi-hop reasoning tasks. Other works have sought to use the language model's internal hidden states to perform implicit reasoning, instead of explicitly producing the chain of thought reasoning steps (Deng et al., 2023; 2024). Geiping et al. (2025). Another line of work sought to fine-tune LLMs, enabling them to reason with explicit continuous tokens. COCONUT (Chain of Continuous Thought) (Hao et al., 2024) operates within the model's hidden state space, eliminating the need for explicit text generation. CODI (Shen et al., 2025) frames the problem as learning to align recurrent hidden states through self-distillation. While these works are promising theoretically, they struggle to generalize to larger models and more challenging benchmarks.

Previous research has established that Large Language Models (LLMs) can generate coherent responses when presented with mixed embeddings as input(Shen et al., 2024; Marro et al., 2025). Building on this, recent frameworks, such as Soft Thinking (Zhang et al., 2025) and Mixture-of-Inputs (MoI) (Zhuang et al., 2025), propose training-free Continuous COT methods for reasoning LLMs. These methods leverage the distribution over the vocabulary at each step to bridge the hidden state output space with the input embedding space. Despite their innovation, these approaches lack a behavioral analysis of the soft decoding procedure. In contrast, our work identifies the Greedy Pitfall in Soft Thinking through comprehensive analysis and introduces random sampling techniques to effectively leverage soft inputs, presenting the first effective Soft Thinking decoding approach.

## 8 CONCLUSION

In this paper, we investigate the Soft Thinking ability of modern LLMs. Contrary to the prevailing assumption that Soft Thinking facilitates the exploration of diverse reasoning paths, we demonstrate that LLMs cannot track multiple reasoning paths simultaneously. Instead, they predominantly rely on the most influential component of the soft inputs during subsequent decoding steps. This reliance hinders the exploration of different reasoning paths and leads to a feedback loop that turns vanilla Soft Thinking into a process resembling greedy decoding, obscuring the advantage of transmitting more information through Soft Tokens. To disrupt this cycle and unleash the potential of Soft Thinking, we propose Stochastic Soft Thinking by introducing controlled randomness into the Soft Thinking Process. Our findings indicate that the Gumbel-Softmax is an ideal randomization approach, as evidenced by both theoretical proof and experimental results. Our paper deepens the understanding of LLMs' latent reasoning abilities through detailed behavior analysis of the generation process and establishes the foundation for further reinforcement learning (RL) training.

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

# A    Dataset Details

**AIME'24&'25**    The AIME'24 and AIME'25 (Mathematical Association of America, 2024) datasets consist of problems from the 2024 and 2025 American Invitational Mathematics Examination (AIME), a highly regarded high school mathematics competition known for its challenging questions. Each dataset includes 30 problems designed to test deep mathematical insight and creative problem-solving.

**MATH500**    The MATH500 benchmark (Cobbe et al., 2021) evaluates the mathematical reasoning and problem-solving capabilities of language models, addressing the growing need for more rigorous assessments as model performance improves. It comprises 500 problems spanning five fundamental mathematical domains: algebra, combinatorics, geometry, number theory, and precalculus. Each problem is crafted to require multi-step reasoning and complex problem-solving, moving beyond basic computation or factual recall.

**AMC'23**    The AMC'23 (mathematics competitions., 2023) dataset contains 40 problems from the 2023 American Mathematics Competitions (AMC), a widely recognized mathematics contest aimed at middle and high school students. The problems emphasize logical reasoning, mathematical creativity, and conceptual understanding, making the dataset a valuable resource for evaluating models on moderately challenging mathematical tasks.

**GPQA-Diamond**    GPQA-Diamond (Rein et al., 2023) is a curated subset of the GPQA benchmark, containing 198 multiple-choice questions across biology, chemistry, and physics. The questions range in difficulty from advanced undergraduate to postgraduate level. This subset includes only those items where both domain experts answered correctly and the majority of non-experts answered incorrectly, ensuring a high standard of quality and discriminative power.

**HumanEval**    HumanEval (Chen et al., 2021) is a benchmark designed to assess the functional correctness of code generated by language models. It consists of hand-written Python programming problems, each paired with a unit test. The benchmark evaluates a model's ability to synthesize correct and executable code from natural language prompts, making it a standard for measuring code generation performance.

**MBPP**    The MBPP (Mostly Basic Python Problems) dataset (Austin et al., 2021) includes 974 crowd-sourced Python programming tasks that cover basic algorithmic and data manipulation skills. Each problem is accompanied by a natural language description and test cases. MBPP is particularly useful for evaluating models on beginner to intermediate-level programming tasks.

**LiveCodeBench**    LiveCodeBench (Jain et al., 2025) is a recent benchmark that evaluates real-time code generation and editing capabilities of language models. It includes a diverse set of programming tasks across multiple languages and domains, emphasizing interactive coding scenarios such as incremental edits, debugging, and code completion. This benchmark reflects practical coding workflows and is designed to test models in dynamic development environments.

