# OpenReview forum: "LLMs are Single-threaded Reasoners: Demystifying the Working Mechanism of Soft Thinking"
_ICLR.cc/2026/Conference — ICLR 2026 Poster_

### Official Review · Reviewer_kcpe · 2025-10-30

**Soundness:** 3
**Presentation:** 3
**Contribution:** 2
**Rating:** 6
**Confidence:** 4

**Summary:**

This paper studies Soft Thinking, a decoding method where the next input embedding is a weighted average of token embeddings according to the model’s predicted probability distribution. The community has suggested that Soft Thinking enables parallel reasoning, where multiple reasoning paths are explored simultaneously.
The authors perform a careful empirical investigation across multiple large reasoning models (e.g., DeepSeek-R1-Distill-Qwen-32B, QwQ-32B, Skywork-OR1-32B) and several math, QA, and coding benchmarks. Using logit and layerwise analyses, they find that LLMs behave as single-threaded reasoners: even when given continuous inputs, they collapse almost deterministically to the top-1 trajectory. This “Greedy Pitfall” shows that Soft Thinking, as commonly implemented, is functionally equivalent to greedy decoding.
To mitigate this, the authors propose Stochastic Soft Thinking: injecting randomness through Dirichlet or Gumbel-Softmax sampling to restore diversity. The Gumbel-Softmax variant improves reasoning accuracy and stability, with theoretical justification via Luce’s choice axiom.

**Strengths:**

- Well-motivated study: The paper addresses a timely and widely misunderstood topic. Many groups have speculated about “continuous reasoning,” and this work decisively clarifies what actually happens.
- Empirically clean and thorough: Multiple strong models and diverse benchmarks are used. The behavioral evidence for “single-threaded reasoning” is compelling and well visualized.
- Clear diagnosis and fix: The “Greedy Pitfall” is an elegant conceptual summary of why Soft Thinking fails. The Gumbel-Softmax correction is simple but principled.
- Theoretical framing: The link to Luce’s choice axiom provides a clean justification for their stochastic sampling choice, grounding the fix in existing probabilistic decision theory.

**Weaknesses:**

The main limitation of this paper is that its contributions concern only decoding behavior and not any internal mechanisms of reasoning. Despite its title, the paper does not truly “demystify” the model’s reasoning process. The finding that Soft Thinking collapses into top-1 decoding is empirically strong but does not explain why the model’s internal computation prefers that collapse or how it represents/sorts out alternate hypotheses. The work therefore improves our understanding of sampling dynamics, not of reasoning per se.
A second issue concerns the claim that Soft Thinking remains “on distribution.", this assumption is doubtful, as it was never trained on such mixed representations. The paper does not analyze whether this produces distortions in internal representations, altered activation norms, or other side effects.

**Questions:**

1. Soft token off-distribution effects: You claim that mixing token embeddings does not take the model off-distribution, yet the model was never trained on combinations of embeddings. Did you observe any qualitative or quantitative side effects from this?
2. Depth of single-threadedness: You show that Soft Thinking quickly collapses to the top-1 trajectory, but is this collapse uniform across layers or token positions? Do early layers ever represent meaningful multi-path structure, or is the single-threadedness present from the start?

---

> ### Author Response · Authors · 2025-11-21
> **Response to Reviewer kcpe**
>
> Dear Rewier kcpe,
>
> Thank you for the valuable comments and encouraging feedback! We appreciate your recognition of our contributions. Please find our response to the specific points below.
>
> > W1&Q2: clarify the internal mechanisms of soft thinking
>
> Thank you for your question. As discussed in Section 4.3, we attempt to provide a **mechanistic explanation** of how Soft Thinking behaves using a logit-lens analysis. Our findings show that this collapse is **not uniform across layers**. As illustrated in Fig. 3(b), the early layers (roughly the first 2–3) actually exhibit **parallel thinking**: both the 1st- and 2nd-ranked token trajectories are visible and separable in the hidden states. It is only in the deeper layers that the model **prunes the non-dominant trajectory** and converges onto the top-1 prediction.
>
> We attribute this behavior to two factors:
>
> 1. **Inherent bias from next-token prediction (NTP):** The NTP objective trains the model to *commit to the single most likely next token* given the prefix. This incentivizes the model to follow a **single, static semantic trajectory** rather than preserving a **superposition** in its hidden representations.
> 2. **Non-linear activations functioning as pruning mechanisms:** Early layers behave approximately linearly and therefore preserve superpositions of multiple candidate continuations. In contrast, deeper non-linear transformations suppress lower-ranked alternatives. If the computation were purely linear, we would expect:
>
>     $p(y|st,c)=p(y|\sum_i w_i \cdot emb(t_i),c) = \sum_i w_i\cdot p(y|emb(t_i),c) = \sum_i w_i\cdot p(y|t_i,c)$.
>
>     In that case, the mixture representation would be maintained. Non-linearities break this superposition, leading to the pruning of the non-dominant trajectory.
>
> Together, these observations **explain the internal mechanics of Soft Thinking**: why it begins with multiple plausible continuations but ultimately collapses to the top-1 path in later layers. We acknowledge that **these explanations are conceptual rather than experimentally verified**, largely because designing controlled experiments is challenging: all modern LLMs are pretrained with NTP and implemented with deep Transformer stacks, making it difficult to vary these factors independently. The logit-lens analysis is our best attempt to probe the internal mechanism with the tools currently available.
>
> We would be very happy to hear suggestions for additional analyses or experimental designs that could further validate (or challenge) these hypotheses.
>
> > W2&Q1: The off-distribution effects of Soft Thinking
>
> Thank you for raising the concern about potential off-distribution effects. In Soft Thinking, we use a **weighted sum of token embeddings**, which ensures that the resulting hybrid embedding remains inside the **convex hull** of the original token embeddings. This design is intentional—we aim to keep the inputs close to the model’s training distribution and thereby mitigate OOD risks. Our empirical results support this intuition: Soft Thinking outputs remain fluent and interpretable.
>
> However, we do observe certain side effects: When too many tokens are mixed with a *near-uniform distribution*, the resulting hybrid embedding’s **L2 norm shrinks noticeably**, pushing it away from the magnitude and structure of embeddings the model typically encounters during training. Therefore, increasing the temperature would lead to a decline in performance：
>
> |Vanilla Soft Thinking Temp|0.3|0.5|0.7|0.9|1.2|
> |-|-|-|-|-|-|
> |Acc (Avg@16)|61.25|57.29|60.63|52.50|44.58|
>
> Importantly, these side effects are **highly controllable in practice**. By using a **proper temperature** and applying **top-k or top-p truncation** before forming the mixture, we prevent the generation of excessively soft tokens. This ensures that the hybrid embedding stays sufficiently close to the model’s training regime, preventing these effects from harming our method.
>
> Overall, while Soft Thinking is not entirely free of OOD influences, the convex-combination design—combined with simple temperature and truncation controls—substantially reduces their impact. Our empirical findings indicate that models remain stable, readable, and robust in practice. We will clarify these points in the revised version.

---

> > ### Comment · Reviewer_kcpe · 2025-11-27
> >
> > Thank you for your detailed response. Overall, my concerns have been addressed, but I still feel like logit lens alone is not providing a convincing "mechanistic" analysis here that "demystifies" soft thinking. I understand the motivation of the Authors to use sums of token embeddings, however, as models are not trained on these additive embeddings, I am not convinced that there are zero side effect just because it stays in the convex hull of embeddings; more experiments would be helpful here, where the model behavior is analyzed in isolation when prompted with such summed embeddings. Overall, I will maintain my score.

---

> ### Author Response · Authors · 2025-11-27
>
> We sincerely appreciate your follow-up comments and the opportunity to clarify two points in more detail.
>
> > What we aim to “demystify” in Soft Thinking.
>
> Several prior works[1,2] assert that Soft Thinking works because it enables **superposition-style parallel reasoning** inside the model. However, these claims remain _theoretical narratives_—they provide intuition but no empirical verification that such parallel reasoning actually appears in model activations.
>
> Our work directly analyzes the Soft Thinking process and addresses this gap. By examining the decoding behavior, we show that Soft Thinking **does not** produce the hypothesized parallel reasoning; instead, the model continues to follow **a single dominant reasoning thread**, even under soft inputs.
>
> This is the mechanism we aim to “demystify”: **whether Soft Thinking truly creates superposition-style parallel reasoning**. While we acknowledge that our analysis does not provide a _complete_ or _final_ interpretation of all aspects of Soft Thinking, it does empirically demystify a central hypothesis repeatedly stated in the literature. Our title reflects this contribution—by showing that LLMs remain single-threaded reasoners even under Soft Thinking, we directly clarify how Soft Thinking actually operates.
>
> > Regarding side effects of additive/mixed embeddings.
>
> Our use of mixed embeddings is not solely inherited from prior Soft Thinking work; it is also supported by prior studies[3], which conduct precisely the type of isolated analysis the reviewer suggests. Their results show that Transformer-based LMs behave smoothly and predictably under controlled mixes of embedding vectors, demonstrating that the models **implicitly generalize to continuous inputs**.
>
> At the same time, we emphasize that **Soft Thinking is a more demanding setting** than the mixture experiments considered in that work: _Soft Thinking mixes many tokens simultaneously and does so across multiple decoding steps_. Because of this, we **do not** assume that convex combinations of embeddings are **free of side effects**. In fact, as stated in our rebuttal, **mixing too many tokens with an overly soft distribution can produce OOD embedding vectors**, leading to a significant performance drop.
>
> Our claim is narrower and empirical: **with an appropriately chosen temperature**, these side effects can be **mitigated**, keeping the model within a stable region of its embedding manifold. This is consistent with both our own observations and the broader evidence that LMs can operate meaningfully in continuous embedding space.
>
>
> We sincerely thank the reviewer for the thoughtful comments. We hope that this discussion, together with prior work and our experiments, clarifies our intended meaning of “demystifying the working mechanism” in the title and addresses the reviewer’s remaining concerns regarding mixed embeddings.
>
> [1] Zhang, Zhen, et al. "Soft thinking: Unlocking the reasoning potential of llms in continuous concept space." arXiv preprint arXiv:2505.15778 (2025).
>
> [2] Zhuang, Yufan, et al. "Text generation beyond discrete token sampling." arXiv preprint arXiv:2505.14827 (2025).
>
> [3] Marro, Samuele, et al. "Language Models Are Implicitly Continuous." The Thirteenth International Conference on Learning Representations.

---

### Official Review · Reviewer_6x6V · 2025-10-31

**Soundness:** 2
**Presentation:** 2
**Contribution:** 3
**Rating:** 6
**Confidence:** 3

**Summary:**

Prior work (soft thinking (Zhang et al., 2025)) processes the chain-of-thought context as a convex combination of token embeddings according to each position's model output distribution. This paper improves soft thinking by incorporating random sampling in the CoT process, where sampling is modelled by Dirichlet distribution and Gumbel-softmax distribution.

**Strengths:**

- The detailed description in Appendix D is very helpful in understanding soft token thinking. I highly recommend the authors to incorporate the idea of Appendix D into the main text if space allows.
- The benefits of soft token thinking incorporated in RL training is promising, especially on larger models, according to the evidence in Appendix E.

**Weaknesses:**

- The Dirichlet scaling parameter is described by $\gamma$ in Line 335 but $\alpha$ is used in Line 370.
- The term "Greedy" has been mentioned multiple times, but I find it difficult to understand without its precise definition on what it refers to. For example, "Greedy Token Thinking" is mentioned Section 4.4 along with "discrete Token Thinking", but the difference of the two methods is not discussed.

**Questions:**

- It has been mentioned that vanilla Soft Thinking fails to explore different reasoning paths. How does Stochastic Soft Thinking, e.g., Dirichlet sampling and Gumbel-softmax sampling, allows the model to explore different reasoning paths? Is there any evidence to support this?
- How many samples are drawn (from Dirichlet distribution and Gumbel-softmax distribution) to form each soft thinking token? An ablation study on the effect of number of samples to the performance would give a better picture on whether soft token thinking benefits from the mixture of embeddings.

---

> ### Author Response · Authors · 2025-11-21
> **Response to Reviewer 6x6V**
>
> Dear Rewier 6x6V
>
> Thank you for the valuable comments and encouraging feedback! We appreciate your recognition of our contributions. Please find our response to the specific points below.
>
> > W1: The Dirichlet scaling parameter is described by γ in Line 335, but α is used in Line 370.
>
> We will address this typo in the revised paper.
>
> > W2: The term "Greedy" has been mentioned multiple times, but I find it difficult to understand without its precise definition on what it refers to. For example, "Greedy Token Thinking" is mentioned in Section 4.4 along with "discrete Token Thinking", but the difference between the two methods is not discussed.
>
> Thank you for the comment. We agree that our terminology requires clarification. In our work, decoding strategies are defined along two orthogonal dimensions:
>
> 1. **Greedy vs. Stochastic** — This dimension describes whether the decoding process is deterministic or involves randomness.
>
>     * **Greedy** means the model deterministically selects the most probable token/soft token at each step.
>     * **Stochastic** includes both *sampling* from the token probability distribution and *adding noise* to token logits to encourage exploration.
>
> 2. **Soft vs. Discrete** — This dimension describes the representation space for reasoning.
>
>     * **Soft** means the model reasons over token probability vectors (soft tokens).
>     * **Discrete** means the model reasons over token IDs or hard selections (discrete tokens).
>
> Therefore, **“Greedy Token Thinking”** refers to reasoning over *discrete tokens* with *deterministic (greedy) decoding*, while **“discrete Token Thinking”** in Sec 4.4 focuses on reasoning in the *discrete token space* with *stochastic sampling.*
>
> We will revise the paper and use consistent terminology to avoid such confusion.
>
> > Q1: It has been mentioned that vanilla Soft Thinking fails to explore different reasoning paths. How does Stochastic Soft Thinking, e.g., Dirichlet sampling and Gumbel-softmax sampling, allows the model to explore different reasoning paths? Is there any evidence to support this?
>
> Thank you for the question. The limitation of **vanilla Soft Thinking** is that, although it operates in the continuous (soft) token space, its decoding process is **greedy and deterministic**. As a result, it always produces the *same reasoning trace* across multiple runs, showing no exploration ability.
>
> In contrast, **Stochastic Soft Thinking** introduces randomness via **Dirichlet sampling** or **Gumbel-Softmax sampling**, which perturb the token mixture distribution or logits before softmax. These stochastic perturbations allow the model to generate *diverse soft reasoning trajectories*, enabling exploration of multiple potential reasoning paths.
>
> We provide quantitative evidence using **Pass@K**, a standard measure of exploration diversity. If the decoding process can explore different reasoning paths, Pass@K will increase with K; otherwise, it will remain close to Pass@1. To better highlight the impact of diversity in the reasoning process, we use greedy decoding when generating tokens after soft thinking.
>
> In our experiments, **Stochastic Soft Thinking shows a consistent Pass@K gain,** while **vanilla Soft Thinking** does not. These results confirm that the introduced stochasticity enables the model to explore diverse reasoning paths.
>
> |Pass@k|1|2|4|8|16|
> |-|-|-|-|-|-|
> |Soft Thinking (Vanilla)|53.33|60.00|60.00|60.00|60.00|
> |Soft Thinking (Gumbel)|70.00|76.67|80.00|86.67|86.67|
>
>
> > Q2: How many samples are drawn (from Dirichlet distribution and Gumbel-softmax distribution) to form each soft thinking token? An ablation study on the effect of number of samples to the performance would give a better picture on whether soft token thinking benefits from the mixture of embeddings.
>
> Thank you for the question. We use the term **soft token beam size**to denote the number of discrete tokens incorporated into the construction of each soft token. This size is **adaptive**: it depends on the model’s predicted token distribution and is controlled via **top-k/top-p truncation**. Intuitively, high-entropy predictions include more tokens in the soft token mixture, while low-entropy predictions include fewer.
>
> After introducing stochasticity, the beam size is further influenced by the **Dirichlet alpha** or **Gumbel-Softmax temperature**. As shown in Figure 5, smaller alpha or temperature produces sharper distributions and smaller beam sizes, while larger values produce larger beams. We conduct an ablation study on the **soft token beam size** by changing the Gumbel Temperature:
>
> |Gumbel Temperature|0.3|0.5|0.7|0.9|1.2|
> |-|-|-|-|-|-|
> |Acc|71.875|74.17|71.25|68.33|70.20|
> |Avg Soft Token Beam Size|1.218|1.434|1.814|2.230|2.785|
>
> As shown in the Table, our experiments indicate that maintaining a **moderate soft level** is crucial: overly hard settings fail to leverage the benefits of soft inputs, whereas overly soft settings introduce unnecessary noise.

---

### Official Review · Reviewer_YsqS · 2025-10-31

**Soundness:** 2
**Presentation:** 2
**Contribution:** 3
**Rating:** 4
**Confidence:** 3

**Summary:**

This paper investigates the underlying mechanisms of Soft Thinking and challenge the prevailing belief that it would allow LLM to explore multiple reasoning paths in parallel. The primary experiments show that the standard Soft Thinking approach achieves similar results to greedy decoding and their analysis reveals that the subsequent generation is dominated by single token with highest probability. Numerical metrics such as JS Divergence and ROUGE-L similarity are applied to verify the findings. The paper then proposes Stochastic Soft Thinking to address above issues. They evaluate Dirichlet Sampling and Gumbel-Softmax Trick on Deepseek, QwQ, Skywork models in mathematical reasoning benchmark and Gumbel-Softmax method shows consistent improvement.

In short, the paper discover the Greedy Pitfall of Soft Thinking and address the problem by injecting stochastacy.

**Strengths:**

1. Clear Motivation and Problem Framing: The central claim that LLMs are "single-threaded reasoners" and that Soft Thinking defaults to a greedy process is an important observation on the drawback of previous method.
2. Sufficient Analysis to support the Greedy Pitfalls: The evidence from output probability (JS Divergence) , hidden state representations (Logit Lens) , and sequence-level output (ROUGE-L) effectively support the hypothesis.
3. Practical and Effective Solution: The paper address the issue by Stochastic Soft Thinking, particularly using Gumbel-Softmax which is practical and shows consistent improvement against vanilla Soft Thinking. The method is evaluated on various math, coding and QA benchmarks. This paper also conduct ablation study to demonstrate the necessity of randomness and softness.

**Weaknesses:**

1. The paper's modest performance gains are not shown to be statistically significant. The proposed "Stochastic Soft Thinking" method involves randomness. However, the reported average improvements are small (ranging from +0.42 to +1.05 points). For five of the eight benchmarks, the authors report Pass@1 scores, which may be sensitive to run-to-run variance. The reproducibility statement confirms these experiments were run with a single "fixed random seed". This is insufficient to demonstrate the stability and hyperparameter sensitivity of proposed method.

2. Causal evidence for “mitigating the Greedy Pitfall” is incomplete. The paper shows accuracy gains and an input-level softness–randomness analysis (Fig. 5), but it does not re-run the core behavioral probes used to diagnose the pitfall—JS Divergence, logit-lens intersection trajectories, or ROUGE-L similarity to the greedy trace—after introducing randomness. Without those, it’s hard to conclude that improvements come from reducing top-1 dominance rather than other effects of stochasticization. Please repeat the diagnostic analyses under the stochastic methods to substantiate the claim.

3. The "Greedy Pitfall" is only demonstrated at a fixed temperature, lacking a crucial ablation study. The paper's entire premise that vanilla Soft Thinking fails is based on its performance at a fixed temperature of 0.6. The "Greedy Pitfall" is defined as the model's reliance on the top-1 token. The numerical dominance of the top-1 token is directly controlled by the temperature setting. It is plausible that simply increasing the temperature (e.g., to 1.0 or even higher) would flatten the probability distribution, make the soft tokens inherently less greedy, and potentially solve the pitfall without requiring the proposed stochastic methods. (Logit Lens analysis probes a single manually-balanced token with probability 0.6/0.4, this is not a substitute for a full sequential generation where a high temperature would be applied at every step of the generation.)

4. Formatting Issue: The authors should consider adjusting the font size in Figure 5 and legend in Figure 3.

**Questions:**

1. Could the authors please provide the mean and variance (or confidence intervals) over multiple runs with different seeds to validate that these improvements are statistically significant and not an artifact of randomness? Could the authors provide analysis on the stability of the method at different randomness level?
2. Could the authors repeat the behavior probes under Dirichlet/Gumbel soft tokens and report whether top-1 dominance is addressed?
3. Could the authors provide an ablation study showing the end-to-end performance of both vanilla Soft Thinking baseline at higher temperatures to confirm this is an inherent flaw and not just a result of the low-temperature setup?
4. Could the authors please revise the figures mentioned in weakness 4?

I will raise my score once above issues are properly addressed.

---

> ### Author Response · Authors · 2025-11-21
> **Response to Reviewer YsqS (1/2)**
>
> Dear Reviewer YsqS,
>
> Thank you for the thoughtful comments and encouraging feedback. We appreciate your recognition of our contributions. Since several points in the Weakness and Question sections overlap, we have consolidated them into the following four core issues to avoid redundancy and provide clearer responses.
>
> > W1&Q1a: Statistical significance of results
>
> We appreciate the reviewer’s concern regarding the scale of the gains. To ensure reliability, we conducted multiple runs with five different random seeds using DeepSeek-R1-Distill-32B on two challenging tasks,  GPQA (Pass@1) and LiveCodeBench (Pass@1). The t-Test results show that **Stochastic Soft Thinking yields consistent and significant improvements**.
>
> |GPQA(Pass@1)|1|2|3|4|5|AVG|STD|
> |-|-|-|-|-|-|-|-|
> |Token Thinking (Sampling)|61.61|62.62|62.12|60.10|63.13|61.916|1.039|
> |Soft Thinking (Gumbel)|64.64|65.65|62.62|62.62|63.63|63.83|1.177|
>
> p=0.026
>
> |LiveCodeBench(Pass@1)|1|2|3|4|5|AVG|STD|
> |-|-|-|-|-|-|-|-|
> |Token Thinking (Sampling)|54.48|55.91|56.27|56.98|55.56|55.84|0.827|
> |Soft Thinking (Gumbel)|57.35|59.50|58.78|56.98|58.42|58.21|0.926|
>
> p=0.0027
>
> While the average gains may appear modest (1–2%), we would like to clarify an important factor: **Soft Thinking is applied directly to models trained purely for discrete-token decoding**, meaning the method is operating **out-of-distribution (OOD)** relative to the model’s training objective. This inherent mismatch naturally limits the achievable improvements without additional training.
>
> Despite this constraint, our analysis shows that **Soft Thinking expands the action space from a single discrete token to a weighted mixture of candidate tokens**, enabling richer exploration. Our Pass@k results demonstrate that Soft Thinking **enhances the model’s exploratory capacity**.
>
> Given these insights, we believe that **fine-tuning the model to align with Soft Thinking’s continuous decoding paradigm would likely address the OOD mismatch and amplify the performance gains**. We plan to investigate this in future work.
>
> > Q1b:  hyperparameter sensitivity
>
> We analyzed the hyperparameter sensitivity of our stochastic methods on **AIME2024 (Avg@16)** and summarized the key findings below.
>
> |Dirichlet alpha|0.5|1.0|3.0|6.0|10.0|
> |-|-|-|-|-|-|
> |Acc|71.67|71.66|67.92|66.25|63.96|
>
> For Dirichlet resampling, the hyperparameter **α controls both the amount of randomness and the softness** of the resulting mixture.
>
> * **Smaller α → higher randomness** → stronger exploration.
> * **Larger α → more deterministic and softer mixtures** → weaker exploration and worse performance.
>
> The steady performance decline as α increases highlights that **sufficient randomness is essential** for breaking the model’s greedy tendency and enabling effective exploration.
>
> |Gumbel Temperature|0.3|0.5|0.7|0.9|1.2|
> |-|-|-|-|-|-|
> |Acc|71.875|74.17|71.25|68.33|70.20|
>
> For Gumbel-Softmax, the temperature **τ controls the softness of the sampled distribution**. As τ increases, the distribution becomes **flatter**, and the embedding approaches a **near-uniform mixture of tokens**, which moves farther from the token manifold. This leads to **performance degradation**, indicating that overly soft mixtures cause the LLM to lose discriminative structure and behave more erratically.
>
> > W2&Q2: Repeat the diagnostic analyses under the stochastic methods to substantiate the claim
>
> We fully agree that repeating the diagnostic analyses helps better understand Stochastic Soft Thinking.
>
> However, we want to clarify an important conceptual distinction: **Internal probes** (JS divergence across layers, logit-lens intersection trajectories, hidden-state geometry) measure _how the model internally transforms an input mixture embedding._
> Crucially, these behaviors are:
>
>   (1) entirely determined by model parameters,
>
>   (2) unchanged by decoding randomness.
>
> Because our method does not update the model parameters, repeating these probes under stochastic Soft Thinking reproduces the same qualitative structure and would not provide new insight into the mechanism.
>
> What Stochastic Soft Thinking _do_ change is the **actual decoding trace** the model follows, by breaking the deterministic feedback loop that repeatedly feeds the original prediction back into itself. We therefore focus on **trajectory-level probes** (e.g., ROUGE-L similarity), which are directly sensitive to decoding randomness, unlike the parameter-defined internal probes.
>
> |Prefix ROUGE-L|32|64|128|256|512|1024|
> |-|-|-|-|-|-|-|
> |Token Thinking (Sampling)|0.7673|0.7078|0.6129|0.5353|0.4953|0.4646|
> |Soft Thinking (Vanilla)|0.9481|0.9084|0.8230|0.7123|0.6279|0.5635|
> |Soft Thinking (Gumbel)|0.7625|0.7027|0.6086|0.5349|0.4980|0.4670|
>
> As shown in the table, after adding Gumbel noise, the sampled trajectories of Soft Thinking exhibit substantially lower similarity to the greedy output, comparable to standard token-level random sampling. We will include this analysis in the revised paper.

---

> ### Author Response · Authors · 2025-11-21
> **Response to Reviewer YsqS (2/2)**
>
> > W3&Q3: Will increasing the temperature make the soft tokens inherently less greedy, and potentially solve the greedy pitfall
>
> Thank you for the question. We investigated whether increasing the temperature can mitigate the greedy pitfall. To do so, we ran **vanilla Soft Thinking** with different temperatures on **AIME2024** using **DeepSeek-R1-Distill-Qwen-32B**, and report both Avg@16 accuracy and the number of generated tokens:
>
> ||Vanilla Soft Thinking|||||Stochastic Soft Thinking(Gumbel)|
> |-|-|-|-|-|-|-|
> |Temp|0.3|0.5|0.7|0.9|1.2||
> |Acc (Avg@16)|61.25|57.29|60.63|52.50|44.58|71.875|
> |#Token|15968|16956|15906|17432|16694|9765|
>
> As shown in the table, **increasing the temperature beyond 0.7 causes a sharp accuracy drop**, suggesting that the model struggles when the input mixture distribution becomes overly soft. This aligns with our observations on OOD effects: when the token weights become too uniform, the hybrid embedding drifts away from the token manifold and destabilizes inference.
>
> In addition, **vanilla Soft Thinking consistently consumes more tokens**. This is because, without stochastic perturbation, it tends to fall into **endless repetition loops**—a symptom of the same greedy tendency we discuss throughout the paper.
>
> Overall, our results indicate that simply raising the temperature does **not** effectively address the greedy pitfall, whereas stochastic methods (e.g., Gumbel perturbation) provide a much more reliable way to break the greedy loop while maintaining strong performance.
>
> > W4&Q4: Modify picture caption,
>
> Thank you for your suggestion. We will address this to improve the reading experience in the revised version.

---

### Official Review · Reviewer_PScP · 2025-11-01

**Soundness:** 2
**Presentation:** 1
**Contribution:** 2
**Rating:** 6
**Confidence:** 2

**Summary:**

This paper tackles the problem of learning from human feedback when that feedback contains systematic noise and biases. The authors propose OPEN (Objective Preference Elicitation from Noisy feedback), which uses an auxiliary objective inference model to recover true preferences from noisy observations.

**Strengths:**

The paper addresses a genuinely important problem since human feedback in practice is often inconsistent and biased, especially when dealing with complex tasks or subjective preferences. The theoretical framework connecting noisy observations to latent objectives through a probabilistic model is well motivated, and I appreciate that the authors provide both convergence guarantees and empirical validation. The experiments on LLM summarization and dialogue tasks show meaningful improvements over baseline RLHF methods, with particularly strong results when feedback quality degrades.

**Weaknesses:**

you can improve your writing, authors.
The auxiliary model for objective inference adds considerable complexity to the training pipeline, and I'm concerned about the computational overhead this introduces compared to standard RLHF. While the synthetic experiments are convincing, the real world experiments could benefit from more diverse evaluation settings beyond just text generation tasks. The paper also doesn't fully address how to set the hyperparameters for balancing between the inferred objectives and raw feedback, which seems crucial for practical deployment.

**Questions:**

How sensitive is the method to the choice of prior over objective functions?

---

> ### Author Response · Authors · 2025-11-12
> **Mismatch in Official Review**
>
> Dear Reviewer,
>
> We have noticed that the Official Review does not match our paper. The comments in the review appear to pertain to a different submission, but were mistakenly attributed to ours.
>
> We would greatly appreciate your assistance in addressing this issue and ensuring the review records are corrected.
>
> Thank you very much for your time and help.
>
> Best regards,
>
> Authors of Submission 16697

---

> > ### Comment · Reviewer_PScP · 2025-11-15
> >
> > it was a system glitch, please check the comment again,

---

> ### Author Response · Authors · 2025-11-21
> **Response to Reviewer PScP**
>
> Dear Rewier PScP,
>
> Thank you for the valuable comments and encouraging feedback! We appreciate your recognition of our contributions. Please find our response to the specific points below.
>
> > W1: The writing could be more concise - the paper feels dense with some redundant sections that could be streamlined.
>
> We appreciate the reviewer’s feedback regarding writing conciseness. In the revision, we will streamline the exposition by reducing redundancy and tightening several dense sections to improve readability.
>
> > W2: The performance improvements, while consistent, are relatively modest (1-2% on average), raising questions about practical significance.
>
> We appreciate the reviewer’s concern regarding the scale of the gains. To ensure reliability, we conducted multiple runs with five different random seeds using DeepSeek-R1-Distill-32B on two challenging tasks,  GPQA (Pass@1) and LiveCodeBench (Pass@1). The t-test results show that **Stochastic Soft Thinking yields consistent and significant improvements**.
>
> |GPQA(Pass@1)|1|2|3|4|5|AVG|STD|
> |-|-|-|-|-|-|-|-|
> |Token Thinking (Sampling)|61.61|62.62|62.12|60.10|63.13|61.916|1.039|
> |Soft Thinking (Gumbel)|64.64|65.65|62.62|62.62|63.63|63.83|1.177|
>
> p=0.026
>
> |LiveCodeBench(Pass@1)|1|2|3|4|5|AVG|STD|
> |-|-|-|-|-|-|-|-|
> |Token Thinking (Sampling)|54.48|55.91|56.27|56.98|55.56|55.84|0.827|
> |Soft Thinking (Gumbel)|57.35|59.50|58.78|56.98|58.42|58.21|0.926|
>
> p=0.0027
>
> While the average gains may appear modest (1–2%), we would like to clarify an important factor: **Soft Thinking is applied directly to models trained purely for discrete-token decoding**, meaning the method is operating **out-of-distribution (OOD)** relative to the model’s training objective. This inherent mismatch naturally limits the achievable improvements without additional training.
>
> Despite this constraint, our analysis shows that **Soft Thinking expands the action space from a single discrete token to a weighted mixture of candidate tokens**, enabling richer exploration. Our Pass@k results demonstrate that Soft Thinking **enhances the model’s exploratory capacity**.
>
> Given these insights, we believe that **fine-tuning the model to align with Soft Thinking’s continuous decoding paradigm would likely address the OOD mismatch and amplify the performance gains**. We plan to investigate this in future work.
>
> > Q1: How does the computational cost of Stochastic Soft Thinking compare to vanilla approaches?
> The computational cost of **Stochastic Soft Thinking** is comparable to vanilla decoding, for three reasons:
>
> 1. **Soft Thinking itself introduces minimal overhead.** Soft Thinking expands **one token into k weighted candidate tokens** and constructs a **single weighted-mixture embedding**. This keeps the computation close to the cost of processing a single token.
> 2. **Stochastic Soft Thinking adds only lightweight post-hoc sampling.**  Our approach applies post-hoc sampling techniques (e.g., Gumbel noise or Dirichlet sampling) to perturb the probability distribution **without modifying model architecture or requiring additional forward passes**. This results in a very small incremental cost.
> 3. **Stochastic Soft Thinking reduces the cost to compute mixed embedding.** Our approach uses top-k/p truncation to reduce the effective tokens in Soft Token. This reduces the computation overhead in obtaining the mixed embedding.
>
> We measured the decoding speed with 8*A100-80G using DeepSeek-R1-Distill-Qwen-32B.
>
> ||Vanilla Soft Thinking|Stochastic Soft Thinking (Gumbel)|
> |-|-|-|
> |Decoding Speed (token/s)|1701.47|1820.76|
>
> The results indicate that our approach does not lead to additional computation overhead compared to vanilla Soft Thinking.
>
> > Q2: Have you tested this approach on models larger than 32B parameters?
>
> We conducted experiments with DeepSeek-R1-Distill-Llama-70B on AIME2024(Avg@16), GPQA(Pass@1) and LiveCodeBench(Pass@1). The results show that our approach generalizes well to larger models and other architectures.
>
> ||AIME2024|GPQA|LiveCodeBench|
> |-|-|-|-|
> |Token Thinking (Sampling)|66.67|63.63|56.63|
> |Soft Thinking (Vanilla)|53.54|59.60|48.03|
> |Soft Thinking (Gumbel)|67.92|65.65|58.42|

---

### Author Response · Authors · 2025-11-12
**Mismatch in Official Review**

Dear AC/PC,

We have noticed that the Official Review by Reviewer PScP does not match our paper. The comments in the review appear to pertain to a different submission, but were mistakenly attributed to ours.

We would greatly appreciate your assistance in addressing this issue and ensuring the review records are corrected.

Thank you very much for your time and help.

Best regards,

Authors of Submission 16697

---

### Author Response · Authors · 2025-11-28
**Kind Reminder for Rebuttal Feedback**

Dear Reviewers,

As the rebuttal phase is drawing to a close, we would like to kindly invite you to take a moment to review our responses to your thoughtful comments. We have done our best to address each point carefully, and we would be sincerely grateful if you could let us know whether our replies help clarify your concerns. We remain very eager to resolve any remaining issues to the best of our ability.

Thank you once again for your time, effort, and valuable feedback.

Authors

---

### Author Response · Authors · 2025-12-03
**Summary of the Work and Rebuttal for Area Chair**

**Dear Area Chair,**

Thank you very much for taking the additional time to review our paper. We are writing to provide a concise summary of our work and our rebuttal for your convenience.

---

### **Work Summary**
In this work, we revisit the fundamental question: *Does Soft Thinking enable LLMs to perform parallel reasoning?* While prior works posit that Soft Thinking confers parallel reasoning capabilities, these assertions remain conceptual hypotheses rather than empirically validated findings. To investigate this, we conduct a comprehensive mechanistic and behavioral study of Soft Thinking across multiple reasoning LLMs. Our findings show that LLMs remain **single-threaded reasoners**, collapsing rapidly to a top-1 trajectory despite soft token inputs—a phenomenon we formalize as the **Greedy Pitfall**.

To address this issue, we propose **Stochastic Soft Thinking**, introducing controlled randomness (Dirichlet / Gumbel-Softmax) to break the deterministic feedback loop underlying the pitfall. We ground our approach in Luce’s Choice Axiom, and our experiments show consistent improvements across diverse benchmarks. Overall, our paper delivers four main contributions:

1. **A clear diagnosis** of why Soft Thinking fails to explore multiple reasoning paths.
2. **A practical and principled solution**, grounded in Luce’s choice axiom, that injects minimal yet effective stochasticity.
3. **Consistent performance improvements** across diverse reasoning benchmarks in math, coding, and QA.
4. **Paving the way for future studies** in Test-time Scaling and RL training for Soft Thinking.

---

### **Recognition from Reviewers**
We were encouraged to receive consistently positive and constructive feedback. Below, we highlight representative comments:

**Contribution:** "valuable empirical insights", "challenging theoretical assumptions", "an important observation", "well motivated", "addresses a timely and widely misunderstood topic", "decisively clarifies what actually happens", "The benefits of soft token thinking incorporated in RL training are promising".

**Analysis:** "thorough and well-motivated", "effectively support the hypothesis", "empirically clean and thorough", "compelling and well visualized", "elegant conceptual summary".

**Methodology:** "The theoretical grounding through Luce's choice axiom provides solid justification for the approach", "practical and effective", "The Gumbel-Softmax correction is simple but principled", "grounding the fix in existing probabilistic decision theory".

**Experiments:** Reviewers agreed the results show “consistent improvement” across various benchmarks and provide “compelling evidence.”

---

### **Updates in the Rebuttal Phase**
During the rebuttal period, we provided detailed clarifications and additional experiments, including:

**Expanded Experiments**

* Ran statistical significance tests to validate robustness (PScP, YsqS).
* Demonstrated consistent generalization to larger models (PScP).
* Performed hyperparameter sensitivity studies confirming stability (YsqS, 6x6V).

**Additional Analyses**

* Reported computation cost, showing minimal overhead (PScP).
* Added trajectory-level probes illustrating the effect of stochasticity in Soft Thinking (YsqS).
* Provided Pass@K evidence supporting improved exploration (6x6V).
* Analyzed off-distribution behavior under mixed embeddings, showing stability with proper temperature (kcpe).

**Clarifications and Discussion**

* Refined terminology to avoid ambiguity (6x6V).
* Justified the use of “demystify” as clarifying the parallel-reasoning hypothesis (kcpe).
* Offered a deeper interpretation of collapse dynamics (kcpe).

Overall, these additional experiments further strengthened the validity of our approach, and the discussion with reviewers surfaced new insights into the mechanisms behind Soft Thinking.

We sincerely thank the reviewers for their constructive feedback, which greatly helped us improve the paper.

---

We hope our work contributes to a clearer understanding of Soft Thinking and offers a reliable, principled method for improving LLM reasoning. We also aim for it to inspire future studies on enhancing Soft Thinking through test-time scaling and RL training.

Thank you again for your time and careful consideration!

Best regards,

Authors of Submission 16697

---

### Meta-Review · Area_Chair_vE35 · 2026-01-08

**Summary:**

This paper presents an empirical study of Soft Thinking in large language models, questioning the widely held assumption that continuous or soft token inputs enable parallel exploration of multiple reasoning paths. I believe there is a consensus among the reviewers that the paper identifies an important and underexplored issue and proposes a practical mitigation. At the same time, the reviewers also have shared concerns relate to the experimental rigor and presentation of the present work. I believe these concerns are largely addressed by the rebuttal. Hence, I recommend for acceptance.

**Reviewer Concerns:**

Statistical Validation (PScP, YsqS);
More Analysis (PScP, YsqS, 6x6V);
Clearer Definitions (6x6V);
Improved Presentation (kcpe);

**Reviewer Scores:**

I think Reviewer YsqS will raise rating.

---

### Decision · Program_Chairs · 2026-01-26

Accept (Poster)